# Racial and Ethnic Disparities in COVID-19 Outcomes: Social Determination of Health

**DOI:** 10.3390/ijerph17218115

**Published:** 2020-11-03

**Authors:** Samuel Raine, Amy Liu, Joel Mintz, Waseem Wahood, Kyle Huntley, Farzanna Haffizulla

**Affiliations:** 1Nova Southeastern University Dr. Kiran C. Patel College of Allopathic Medicine, Davie, FL 33328, USA; al2059@mynsu.nova.edu (A.L.); jm4719@mynsu.nova.edu (J.M.); ww412@mynsu.nova.edu (W.W.); kh1607@mynsu.nova.edu (K.H.); 2Department of Internal Medicine, Nova Southeastern University Dr. Kiran C. Patel College of Osteopathic Medicine, Davie, FL 33328, USA

**Keywords:** COVID-19 pandemic, social determination of health, disadvantaged populations, occupational risk, race, gender, poverty, intersectionality

## Abstract

As of 18 October 2020, over 39.5 million cases of coronavirus disease 2019 (COVID-19) and 1.1 million associated deaths have been reported worldwide. It is crucial to understand the effect of social determination of health on novel COVID-19 outcomes in order to establish health justice. There is an imperative need, for policy makers at all levels, to consider socioeconomic and racial and ethnic disparities in pandemic planning. Cross-sectional analysis from COVID Boston University’s Center for Antiracist Research COVID Racial Data Tracker was performed to evaluate the racial and ethnic distribution of COVID-19 outcomes relative to representation in the United States. Representation quotients (RQs) were calculated to assess for disparity using state-level data from the American Community Survey (ACS). We found that on a national level, Hispanic/Latinx, American Indian/Alaskan Native, Native Hawaiian/Pacific Islanders, and Black people had RQs > 1, indicating that these groups are over-represented in COVID-19 incidence. Dramatic racial and ethnic variances in state-level incidence and mortality RQs were also observed. This study investigates pandemic disparities and examines some factors which inform the social determination of health. These findings are key for developing effective public policy and allocating resources to effectively decrease health disparities. Protective standards, stay-at-home orders, and essential worker guidelines must be tailored to address the social determination of health in order to mitigate health injustices, as identified by COVID-19 incidence and mortality RQs.

## 1. Introduction

The high incidence of novel coronavirus disease 2019 (COVID-19) in the United States is concerning; there have been approximately 8.2 million confirmed cases and nearly 220,000 deaths [1]. Blumenshine et al. described an imperative need for policy makers at all levels to consider socioeconomic and racial/ethnic disparities in pandemic planning [2]. They developed a conceptual framework to illustrate the three levels by which these disparities might result in unequal rates or illness or death: disparities in exposure, in susceptibility to contracting, and in treatment of the disease [2]. Therefore, identification of social vulnerabilities and disparities may allow for development of focused and culturally sensitive interventions to limit transmission and mortality caused by any pathogen, including COVID-19.

In the United States, the increasing COVID-19 burden has exposed longstanding socioeconomic and health disparities. Differences in COVID-19 incidence and mortality vary dramatically, and are apparent between age groups, geographic regions, and races. Studies find that Black people are dying from COVID-19 at higher rates than any other race or ethnicity [3,4]. Examination of the forces which influence the social determination of health may help explain these differences and spur innovation and policy changes to address health injustices.

The theory of social determination of health proposes that social dynamics, in addition to risk factors, dictate the health of individuals as a result of the inequalities created [5]. The World Health Organization (WHO) defines social determination as “the forces and systems shaping the collective conditions in which people are born, grow, work, live, and age, as well as the conditions of their daily lives [6]”. Individual social dynamics give rise to the risk factors affecting COVID-19 exposure, susceptibility, and differences in treatment-seeking behavior or access to care. Dismantling unjust systems which lead to inequities related to the social determination of health is the foundation of health care justice. Equitable allocation and access to resources allows all individuals to attain their highest level of health.

Studies indicate that chronic disease outcomes and social mobility are more influenced by zip code than immediate family resources [7]. This suggests that regional differences in community exposure to poverty, food insecurity, oppression, abuse, racism, and environmental hazard exposure contribute to the development of long-term stress, which, in turn, results in physiological maladaptation and perpetuates adverse health outcomes [8]. There is a well-documented correlation between social dynamics and chronic diseases such as diabetes, hypertension, schizophrenia, depression, and asthma [6]. However, there are limited data examining the efficacy of public policies to mitigate the social determination of health in the COVID-19 pandemic.

Between March and June 2020, stay-at-home orders, mandatory face coverings, and social distancing guidelines were instituted across the US. These policies have affected employment, travel, behavior, and mental health [9,10,11]. However, it is unclear whether these policies have equal impact on risk of exposure, susceptibility to contraction, and access to treatment among all people. In this study, we aim to evaluate national- and state-level racial and ethnic disparities in preliminary COVID-19 cases, outcomes, and identify potential factors that contribute to any disparities.

## 2. Materials and Methods

The protocol for our cross-sectional analysis of public-use data was approved by the institutional review board at Nova Southeastern University. There was no requirement for informed consent as the data were de-identified and made publicly available. To maintain adequate standards of assessment, the Strengthening the Reporting of Observational Studies in Epidemiology (STROBE) guidelines were used in crafting this protocol [12].

There were 2 objectives to the statistical analysis: (1) to determine racial/ethnic trends of COVID-19 incidence relative to the US population, and (2) to evaluate the racial/ethnic distribution of COVID-19 related mortality relative to the US population.

### 2.1. Data Sources

We analyzed data from The COVID Tracking Project’s Racial Data Tracker, compiled by the Center for Antiracist Research at Boston University, which aggregates state-level COVID-19 reporting and tracking databases (Appendix A) [13]. The numbers of confirmed COVID-19 cases and mortalities were evaluated by race and ethnicity on an individual state level. We compared the proportion of American Indian or Alaskan Native (AIAN), Black, Hispanic/Latinx, Native Hawaiian or Pacific Islander (NHPI), Asian, and White cases and mortality attributed to COVID-19 to the corresponding proportions in the state populations, by using self-reported national and state demographic data from the United States Census Bureau’s 2018 American Community Survey (ACS) 5 year averages (Appendix A) [14].

Data on White, Black, Hispanic/Latinx, Asian, AIAN, and NHPI people were evaluated. We excluded the multiple race/ethnicity category from our analysis, as there was insufficient data for analysis. COVID-19 incidence data classified by race and ethnicity were available in 46 of 50 US states. Racial and ethnic data from Louisiana, New York, North Dakota, and Rhode Island were not available and were excluded from our analysis. Cases and deaths where race or ethnicity were unknown were also excluded. The data were restricted to reports from 1 March to 14 June 2020, when public policy stay-at-home orders were in effect, regardless of their enforceability [15].

### 2.2. Statistical Analysis

A representation quotient (RQ) was utilized. RQs were defined as the proportion of a particular subgroup in the total number of COVID-19 cases or mortality relative to the corresponding estimated proportion of that subgroup in the United States population. For example, COVID-19 incidence RQ for White people would be given by $RQ=\frac{\%\;White\;incidence}{\%\;White\;population}$. An RQ greater than 1 indicates that a subgroup is over-represented, and a RQ less than 1 indicates that the subgroup is under-represented. The magnitude of the RQ can be used to evaluate the scale of the under- or over-representation. For example, a COVID-19 death RQ of 1.5 indicates that the subgroup’s representation in COVID-19 mortality is 50% greater than the subgroup’s representation nationally. The RQ is a useful method in disparity analysis as it contextualizes findings. For example, a reported influenza incidence of 70% for subpopulation X and 30% for subpopulation Y appears to indicate that subpopulation X is overly affected by influenza. However, this does not consider that subpopulation X comprises 90% of the total population, and subpopulation Y only 10%. The disproportionate effect of influenza on subpopulation Y is not apparent until it is compared to the size of Y relative to the size of the general population. Using RQ to compare the effect clearly illustrates the disparity; subpopulation X RQ = 0.778, subpopulation Y RQ = 3.0. Choropleth maps were generated from the state-level RQs using the internet-based data packing platform, Displayr (NSW, Sydney, Australia) [16].

After calculation of incidence and mortality RQ by race and ethnicity, Tukey’s test for post-hoc analysis was completed with an ANOVA to evaluate statistically significant (*p* < 0.05) differences between the average state RQs between all racial/ethnic groups.

## 3. Results

Between 1 March and 14 June 2020, the number of confirmed COVID-19 cases rose from 69 individuals to 2.07 million in the United States. The total confirmed deaths during this same period increased from 1 individual to 115,436 [13].

### 3.1. Racial/Ethnic Trends in COVID-19 Incidence

#### 3.1.1. COVID-19 Incidence in the US

Among all racial/ethnic groups, White people had the highest number of confirmed cases, approximately 29.56% of total cases. However, the case RQ for White people was only 0.484. This indicates that, for White people, COVID-19 case representation is approximately 48% of the group’s representation in the total US population. Non-White, non-Black Hispanic/Latinx accounted for 25.58% of the confirmed cases in the US and had a RQ of 1.437. This indicates that the representation of Hispanic/Latinx in COVID-19 cases was 43% greater than the group’s representation in the US population. AIAN, NHPI, and Black people all had RQs >1; 1.303, 1.115, and 1.278, respectively. The case RQ for Asian people was 0.509. US RQ data for all racial and ethnic groups evaluated can be found in Appendix B, Table A1. These RQ values suggest that there are racial and ethnic disparities in COVID-19 case incidence on a national level, during the 1 March to 14 June 2020 period. Table 1 summarizes the national RQ data for COVID-19 cases and mortality by race/ethnicity. Significant differences between race/ethnicity incidence RQs from Tukey’s test for post-hoc analysis can be seen in Appendix B, Table A2.

#### 3.1.2. COVID-19 Incidence by State

Within racial and ethnic groups, RQ variability by state was observed. Among White people (Figure 1A), the average RQ = 0.560 but ranged from 0.114 in Texas to 0.982 in Hawai’i. Despite this variability, no RQ > 1 was observed for White people in any state, indicating that White people had lower COVID-19 case representation relative to representation in the state population across all states.

In contrast, the average RQ among Black people was 2.129 (Figure 1B), and ranged from 0.248 in Texas to 17.571 in Maine. Only 11 states had a case RQ < 1 for the Black population. The average state-level case RQ for Asian, AIAN, NHPI, and Hispanic/Latinx people was 1.026, 1.341, 0.858, and 2.182, respectively. Figure 1A–F, below, shows choropleth maps of the RQs from the race/ethnicity-specific analysis by state.

### 3.2. Racial/Ethnic Trends in COVID-19 Mortality

Racial and ethnic data were available for COVID-19-related mortality in 45 out of the 50 US states. Racial and ethnic mortality data were unavailable for Hawai’i, Montana, Rhode Island, South Dakota and North Dakota.

#### 3.2.1. COVID-19 Mortality in the United States

White people accounted for the largest proportion of confirmed mortality by race, approximately 49.4% of total US deaths. However, the case RQ for White people was only 0.808. This indicates that the White population’s representation in COVID-19 deaths is 80.8% of the White representation in the total US population. Black people accounted for only 22.04% of the total number of confirmed deaths in the US, but had a RQ of 1.792. indicating that Black people account for a larger portion of COVID-19 deaths relative to their representation in the US population. Asian, AIAN, and NHPI people each had RQs less than 1; 0.742, 0.829, and 0.352, respectively (Table 1). These results indicate that there is a significant disparity in COVID-19 deaths between the Black population and all other racial/ethnic groups, during the 1 March to 14 June 2020 period. Significant differences between race/ethnicity mortality RQs from Tukey’s test for post-hoc analysis can be seen in Appendix B, Table A3.

#### 3.2.2. COVID-19 Mortality by State

In state-level analysis, the average mortality RQ for COVID-19 deaths among White people (Figure 2A) was 0.808, ranging from 0.295 in the District of Columbia to 1.160 in Idaho and Oklahoma. In contrast, the average mortality RQ for Black people (Figure 2B) was 1.52 and ranged from 0.368 in Texas to 4.409 in Kansas. Only 7 states had a mortality RQ < 1 for Black people. Compared to all other racial/ethnic groups, Black people had the largest state-average COVID-19 death RQ. The mortality RQ for American Indians and Native Alaskans (Figure 2E) was greater than 3 in many of the states across the southwest and accounted for the population with the highest mortality RQ in any state (RQ = 22.72 in WY). The average mortality RQ for Asian, NHPI, and Hispanic/Latinx people was 0.711, 0.630, and 0.707, respectively. It is important to note that the Asian, NHPI, and Hispanic/Latinx groups had mortality RQs > 1 in multiple states. Figure 2A–F shows choropleth maps for the deaths RQ from the race/ethnicity-specific analysis on a state-by-state level. Appendix B contains the numeric score of racial/ethnic death RQs.

## 4. Discussion

The United States has received global criticism on its response to the COVID-19 pandemic [17,18,19]. The initial lag in testing combined with the development of faulty test kits severely impaired COVID-19 containment in the United States [20]. In June, there was a significant rise in the number of new COVID-19 cases which coincided with the reopening of many states, as well as increased politicization of stay-at-home orders and the subsequent reduction in public health measures such as mask wearing [21]. The following discussion will evaluate the factors contributing to COVID-19 inequality by the three levels of disparity, described by Blumenshine et al.: (1) disparities in exposure, (2) in susceptibility to contracting, and (3) in health care-seeking behaviors and treatment of the disease [2].

### 4.1. Disparities in Exposure

#### 4.1.1. Essential Worker Policies

The intent of stay-at-home orders and essential worker policies was to limit social gatherings within crowded workplaces and reduce person-to-person transmission. The US Cybersecurity and Infrastructure Security Agency (CISA) published categorized the following industries in an advisory memorandum as critical infrastructure sectors, and therefore as essential work: medical and health care, telecommunications, information technology systems, defense, food and agriculture, transportation and logistics, energy, water and wastewater, law enforcement, and public works [22]. This served as a guideline for individual states in the issuance of stay-at-home orders.

According to the Center for Economic and Policy Research’s 2019 analysis of the essential workforce, over 31 million Americans are deemed essential. Approximately 50.9% of these workers are in the health care sector, 20.6% in food and agriculture, and 7.2% in transportation and delivery [23]. These essential workers risk increased exposure to COVID-19. Low-income and people of ethnic and racial minorities are more likely to work in sectors that remained open [24,25].

In the United States, more than 36% of essential workers are non-White [23]. Furthermore, non-White workers are over-represented in the ten largest health care occupations, in childcare, and in public transportation employment in the US [23]. The large number of non-White essential workers, and potential increased risk of SARS-CoV-2 exposure among essential workers, may contribute to racial disparity in COVID-19 incidence. For example, in California, where Hispanic/Latinx people make up 39.6% of the essential workforce, the case RQ for Hispanic/Latinx in California was 1.031 compared to 0.341 for White residents. The majority of Hispanic/Latinx essential workers in California are non-health care workers, which may also increase risk due to less availability of protective equipment and workplace precautions. However, there is some variability between states, and exposure of essential workers cannot account for all racial and ethnic disparities observed in COVID-19 case incidence and death rate. For example, essential worker demographics in Utah closely resemble state demographics, yet the case and death RQ for White residents is <1 and case and death RQs for Black, Asian, AIAN, NHPI, and Hispanic/Latinx residents are all >1.

There are distinct differences in the proportion of workers who are eligible to work from home by race and ethnicity. The US Bureau of Labor Statistics reported that in the period 2017–2018, 42 million wage and salary workers (29%) had the option to work from home. Of these, 36 million reported working from home at least occasionally [26]. While 26% of White people and 32% of Asian people were able to work from home, only 13% of Hispanic/Latinx and 18% of Black people had the option [26]. There was also an increased likelihood of eligibility to work from home associated with advanced education, indicating that workers with lower levels of education may be more likely to work in an essential sector [26]. Lower levels of educational attainment are also associated with higher unemployment rates and lower median weekly earnings [26]. The intersection of education and income inequality may result in greater pressure to work, regardless of personal safety. Forty-seven percent of those over the age of 25, with at least a bachelor’s degree worked at home, compared to only 9% of those with a high school diploma and 3% of those with less than a high school diploma [26]. Individuals who can work from home may have less of an interpersonal and occupational risk for virus exposure.

#### 4.1.2. Transportation

Personal transportation methods vary widely across the US. In rural areas, Americans rely heavily on personal vehicles to travel to their place of employment [27]. Conversely, urban areas with greater access to public transportation (such as buses, trains, and subways) may have lower rates of car ownership [27]. In the United States, only 7% of White households do not own a car, compared with 24% of Black households, 17% of Hispanic/Latinx households, and 13% of Asian households [27]. Without access to a vehicle, a larger portion of minority populations are reliant on public transportation services. In US urban areas, Black and Hispanic/Latinx residents comprise 44% of public transport users (62% of bus, but only 35% of subway, and 29% of commuter rail) [28]. Dependence on public transportation increases interpersonal proximity and, subsequently, risk for exposure to COVID-19 [29]. In a study conducted in New York City, a greater proportion of essential workers residing in a neighborhood was independently associated with higher rates of subway use during the pandemic [30]. There was also a positive correlation between subway use and lower median income, as well as areas with a greater percentage of non-White and/or Hispanic/Latinx residents. However, these correlations weaken when controlled for proportion of essential workers, suggesting that essential work is the driver of subway use in lower socioeconomic neighborhoods [30].

### 4.2. Disparities in Susceptibility

#### 4.2.1. Housing and Environment

Early reports from Japan show COVID-19 incidence rates are heavily influenced by population density [31]. Density driven transmission can also be seen on the micro-level, such as within a crowded household. The ACS reports that approximately 1 in 5 families reside in a multigenerational home environment. This differs dramatically by race and ethnicity; only 3.7% of non-Hispanic White households are multigenerational, compared to 9.5% of Black, 10.7% of AIAN, 9.4% of Asian, 10.3% of Hispanic/Latinx, and 13% of NHPI households. Essential workers returning home to multigenerational households may be putting those in their homes at increased risk of contracting COVID-19 [14]. Children have been found to be predominantly asymptomatic carriers of the virus, which may increase the risk of transmission to other household members [32]. This poses a significant threat to elderly members of multigenerational households, who are at higher risk of COVID-19 morbidity and mortality [13,33,34].

Limited access to potable water or indoor plumbing on Native American reservations makes it difficult to follow handwashing guidelines. The study by Rodriguez-Lonebear et al. found that among AIAN, increased likelihood of COVID-19 was associated with the lack of indoor plumbing [35]. The association between infection and indoor plumbing is unsurprising, as handwashing has been heavily purported as an effective means of protection against the infection [36].

#### 4.2.2. Racial Discrimination and Systemic Oppression

Socially assigned race has been shown to have greater impact on health than self-reported race [37]. These systems result in disproportionate allocation of resources such as education, employment, housing, and health services, among others. They also lead to the unequal distribution of burden of disease, incarceration, and pollution. For example, there is a higher prevalence of hypertension, diabetes, obesity, and cardiovascular disease among Black people, which contributes significantly to COVID-19 vulnerability [38]. There is a correlation between perceived racial discrimination and hypertension, strengthened by Black race [39].

Beginning in May, Black Lives Matter and other social justice movement demonstrations were held nationwide. These public demonstrations were sparked by the murders of unarmed Black people by police, including Breonna Taylor and George Floyd, among others. Public health officials initially raised concerns that these protests could potentially increase the spread of COVID-19. However, it is unclear whether these large-scale protests have directly impacted COVID-19 outcomes. A notable increase in COVID-19 incidence occurred shortly after the protests began, but this also coincided with the “reopening” of many states and a marked increase in overall social activity. Interestingly, studies have shown that cities which had large protests actually had a net increase in social distancing behavior [40].

Many argue that racial injustice, and the longstanding mistreatment of Black people by the criminal justice system, is a public health crisis and necessitates these protests. Indeed, the incarceration of Black people occurs at a rate 5 times higher than their White counterparts [41]. Black and Hispanic/Latinx communities face similar disparities in incarceration rate as in COVID-19 incidence and mortality. The ratios of incarcerated Black people to White people in Minnesota, Iowa, Wisconsin, New Jersey, and Connecticut are all >8, and the Hispanic/Latinx to White ratios are >1.2 [41]. In New Jersey, where the ratio of Black to White inmates is >8, almost 15% (2892 individuals) of the incarcerated population tested positive for COVID-19 [42].

Structural racism contributes to disproportionate rates of incarceration; there are higher rates of incarceration among Black, Indigenous, and Hispanic/Latinx individuals, particularly those of low socioeconomic status and those with mental illness, than their White counterparts [43]. Moreover, incarceration makes it fundamentally impossible to practice social distancing; close contact is nearly unavoidable when prisons are overcrowded. Familial and social disruption as a result of incarceration may also predispose those individuals to poverty and poorer long-term health outcomes, increasing risk of chronic and acute illness. Although incarceration may not directly increase COVID-19 risk, it may compound other factors which contribute to worse COVID-19 outcomes.

### 4.3. Disparities in Treatment and Health-Promoting Behaviors

#### 4.3.1. Insurance

Almost 30 million Americans do not have health insurance coverage [44]. Although testing is often free, individuals may choose not to seek testing services for concerns over the cost of treatment. In a study of 14,036 people receiving COVID-19 testing, only 6.7% were Black and 19.1% Hispanic/Latinx, compared to 48.3% who were White [45]. Many Americans receive insurance plans through their places of employment. However, insurance coverage may have been impacted by COVID-19 as almost 13% of the workforce has faced job termination or employment reduction between February and May of 2020 [46].

Health insurance programs may be expensive, inaccessible for many individuals, and have complex policies for covered expenses. Even when enrolling in marketplace plans with tax credits, only 69% of individuals had premiums of <$100 per month [47]. This barrier to accessibility widens the health care gap for low-income individuals. Insurance and socioeconomic data of COVID-19 patients are not readily available in regard to race and ethnicity to evaluate impacts in the identified disparities, but opportunities exist to improve insurance coverage to the most in need. Proponents of single-payer health care in the US cite this failure of the Affordable Care Act (ACA) to reduce individual cost [48]. They argue that while the ACA has increased the number of insured, there are still 25 million uninsured US residents; a single-payer system would lower administrative costs and increase accessibility of health care. Recent economic-benefit models of a public option of Medicare and Medicaid expansion suggest that elective coverage to lower-income, uninsured Americans has variable federal cost savings [49,50]. While all conclusions from these models were reliant on a specific set of assumptions surrounding the health care market, opportunities for a more inclusive and just health system should not be dismissed.

#### 4.3.2. Access to Treatment

Access to testing and treatment varies across the US. Of greatest concern is affordability of care. Several states, such as Michigan, Wisconsin, and New York, have reported that the zip codes most impacted by COVID-19 are also the poorest [51]. In New York City, the highest concentration of COVID-19 cases was found in Brooklyn, Queens, and the Bronx, where there are larger Black, Hispanic/Latinx, and Asian populations. Relatively fewer cases have been reported from the predominantly White, upper-class neighborhoods of Manhattan [51].

In an effort to expand services to rural health clinics, the US Department of Health and Human Services allocated $225 million dollars to support expansion of COVID-19 testing, telemedicine services, and notification/information services [52]. Using these funds, the Food and Drug Administration developed multilanguage COVID-19 health information flyers supporting up to 40 different languages. However, website information is limited to English and Spanish, with only a few pages available in six other languages [53]. Therefore, residents with low English proficiency and/or literacy may have difficulty accessing accurate information on symptom identification, testing, and treatment. Furthermore, internet access is required to access online health information, which may preclude a large, mainly low-income proportion of the population.

In 2017, the Federal Communications Commission identified approximately 25 million Americans that do not have internet access, of which, approximately 19 million live in rural areas [54]. Further, 157 million do not operate the internet at broadband speeds, a standard for most Americans since 2000 [54]. Telemedicine has become increasingly common over the course of the pandemic. Remote medical appointments allow patients to receive health care services during the pandemic. Unfortunately, this service is inaccessible to those without internet connectivity. This lack of service may further exacerbate health care access disparity in rural and underserved areas of the US.

#### 4.3.3. Health and Nutrition

Insufficient access to nutritious foods, low health literacy, and lack of culturally appropriate health education compound risk of hypertension, diabetes, and obesity—all of which are risk factors for COVID-19 morbidity and mortality [55,56,57]. Access to healthy, affordable foods is limited in racially segregated and deprived neighborhoods [58]. While food insecurity is prevalent nationwide and affects all races/ethnicities, a higher incidence is found along the Southern Belt as well as areas in the Midwest [59,60]. Black individuals are 1.8 times more likely to have diabetes than their White counterparts, which may be partially attributed to food insecurity [61]. Black individuals are more likely to live in lower-income neighborhoods with greater food insecurity as a result of racialized segregation [61]. Native American and Alaskan Natives are also disproportionately impacted by disparities in health and nutrition. The average age of mortality due to diabetes is 68.2 years among AIAN individuals, compared to 74.6 years among their White counterparts [61]. The social determination of health as a result of food insecurity and related metabolic diseases closely corresponds with our RQ findings. Higher average COVID-19 incidence RQs are seen in the Black and Hispanic/Latinx populations along the Southern Belt (1.591, 1.812), where a higher incidence of food insecurity has been reported [60].

Vitamin D deficiency has also been found to be associated with increased risk for COVID-19, further worsening the outlook for the undernourished [55]. Studies show that living in northern US states may account for lower levels of Vitamin D across all races/ethnicities due to lower average daily sunlight hours, especially during the winter [62,63]. Vitamin D levels, primarily synthesized by exposure to sunlight, are lower on average in people with darker shades of skin. This may lead to increased COVID-19 vulnerability [64]. During our study period, there was a general pattern of increasing incidence and death RQ with increasing latitude, regardless of race or ethnicity. However, it is unclear whether these trends are a result of lower levels of vitamin D or other geographic variances. Further investigation of the relationship between vitamin D levels, geographic location, and COVID-19 incidence and mortality is warranted.

## 5. Limitations

We suspect that COVID-19 incidence and mortality may be under-reported given the initial difficulty in testing. Studies indicate that, after controlling for COVID-19 deaths, the overall mortality rate has increased during the March–May 2020 period compared to average mortality rates for the past 10 years [64]. This suggests that US coronavirus mortality may be under-reported. Additionally, asymptomatic individuals may not even seek evaluation.

Another limitation to this study was the variability in state-level data. There is no standard method for data collection by race, sex, and ethnicity between states. For example, some states categorized more than 25% of reported cases as “other” race or ethnicity. This aggregation of data may obscure other disparities, yet to be defined, and may have led to underestimation of incidence and mortality RQs. Additionally, our study only evaluated state-level incidence and mortality, which may have masked greater disparities between counties and cities within states.

During the study period, 8 states had fewer than 100 COVID-19 deaths, some with less than 40. This may overestimate racial and ethnic disparities due to a low sample size. Cases and mortality continue to fluctuate nationwide and as more data become available, repeat analysis may be warranted using an extended time frame.

## 6. Conclusions

National data may mask racial and ethnic disparities in COVID-19. Microcosmic data, from city or state-level reporting, better illustrate the disparate health outcomes during the pandemic. The evidence suggests that there is significant disparity in COVID-related health outcomes by race and ethnicity regarding representation by state. We suspect that these disparities would be even more apparent at the county and city level and this warrants further investigation. There is a need for public reporting of disaggregated COVID-19 data by county and zip code. Transparency of local data would allow for greater precision in allocation of resources and establishment of effective policy changes to disrupt the social determination of poor health outcomes, moving the United States towards the goal of health justice.

Infectious disease, including COVID-19, does not selectively affect individuals based on race. However, the social dynamics which perpetuate racial, economic, and environmental disparity create a system of injustice which sustains health inequality, thereby resulting in disparate susceptibility to infection, morbidity, and mortality among marginalized communities. It is an exercise in futility to search for any single element that predisposes people of color, in particular Black people, to adverse COVID-19 outcomes. Instead, we need to explore the intersectionality of inequity and the social determination of health. Therefore, we believe that racial and societal disparities are not only cultural and economic issues, but also an issue of public health and social justice.

Many of these inequalities are longstanding and rooted in our society’s infrastructure. The dismantling of systems of oppression which drive the ongoing health injustice epidemic is requisite for a reduction in health disparities and relief of COVID-19 disease burden. Limiting and reducing mortality require identification of disparity dynamics in our communities and addressing the social determination of these vulnerabilities through specific, intentional interventions. Although systemic change may take time, we hope that health injustices, as exemplified by the COVID-19 pandemic, serve to spark action and create momentum towards achieving social justice and eliminating health disparities.

## Figures and Tables

**Figure 1 ijerph-17-08115-f001:**
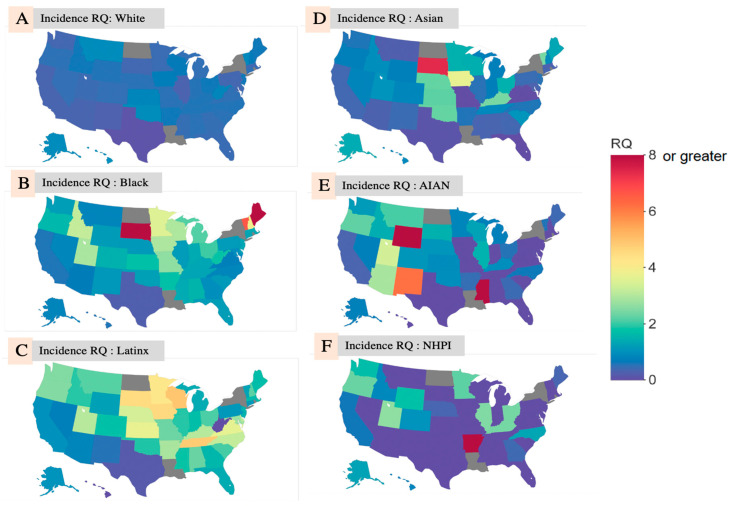
Choropleth Maps of COVID-19 Incidence RQ by State among (**A**): White people; (**B**): Black people; (**C**): Latinx people; (**D**): Asian people; (**E**): AIAN; (**F**): NHPI.

**Figure 2 ijerph-17-08115-f002:**
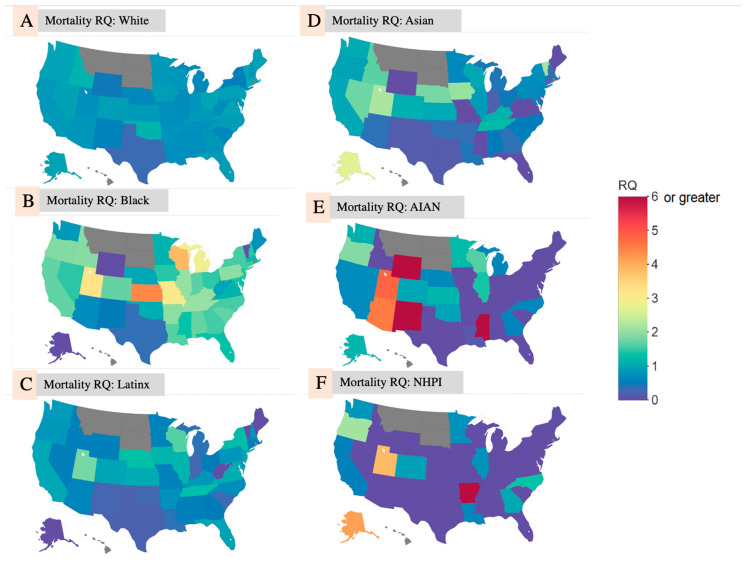
Choropleth Maps of COVID-19 Death RQ by State among (**A**): White people; (**B**): Black people; (**C**): Latinx people; (**D**): Asian people; (**E**): AIAN; (**F**): NHPI.

**Table 1 ijerph-17-08115-t001:** US Incidence and Morality Representation Quotients (RQ) for COVID-19 by Race and Ethnicity.

Race or Ethnicity	US Population	Incidence ^1^	Mortality ^2^	Incidence RQ	Mortality RQ
White	61.10%	29.56%	49.40%	0.484	0.808
Latinx	17.80%	25.58%	14.87%	1.437	0.835
Black	12.30%	15.72%	22.04%	1.278	1.792
Asian	5.40%	2.75%	4.00%	0.509	0.742
Multiracial	2.40%	0.59%	0.20%	0.246	0.082
AIAN	0.70%	0.91%	0.58%	1.304	0.829
NHPI	0.20%	0.22%	0.07%	1.116	0.352
Unknown ^3^	0.20%	25.58%	6.20%	0.484	0.808

^1^ LA, ND, and NY did not publicly report case data by race/ethnicity; ^2^ MT did not publicly report death data by race/ethnicity; ^3^ unknown racial and ethnic data in The COVID Tracking Project represents a significant portion of COVID-19 cases and deaths due to lack of standardized data collection methods.

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
