# Peer review of "Racial and Ethnic Disparities in COVID-19 Outcomes: Social Determination of Health"

_ijerph, 2020, doi:10.3390/ijerph17218115_

Round 1

Reviewer 1 Report

The study by Raine et al documents the racial discrepancies in Covid-19 incidence in the United States of America. The authors use demographic data reported in 46 states in combination with infection and death rates associated with Covid-19. This is an important study investigating the racial differences in disease burden and that are linked to differences in socio-economic factors between white and non-white Americans.

The authors use a simple method comparing proportion of Covid positives compared to proportions of general population for different ethnicities. They show an overrepresentation in Covid cases for non-whites, particularly the black population. However, they found some variations across states. The Introduction, methods and results are well written and easy to follow; however, the discussion is disproportionally long and complicated. I believe the manuscript will benefit from an overall simplification of the text.

Major comments:

The authors use a representation quotient that is the proportion of Covid cases for an ethnicity divided by the proportion of that ethnicity overall. While this does not seem statistically incorrect it is not clear to me why the authors decided to choose this method rather than comparing incidence rates across ethnicities? I believe, a discrepancy in positive cases for individual ethnicities should be represented in the overall incidence rate. For example, incidence of Covid-19 in white is Xy, in black is yte, …

I think it would be useful to explain why the documented method was chosen.  

Did the authors compute any statical power analysis? This should be stated otherwise the differences are non-significant.

The discussion is a bit confusing. It states a lot of information regarding differences in socio-economic factors between different ethnicities. Some of these might be better suited in the introduction, for example section 4.1. While these are important facts it is not clear to me which factors best explain the discrepancies between states.

Minor comments:

Table 1.  I think it would be better to order the ethnicities according to % of population (high to low). Also, if the column header states % it does not need to be added to the individual values below.

Lines 149ff. “Black people had the largest average COVID-19 death RQ. The death RQ for American Indians and Native Alaskans was greater than all groups except for Black people.”

This is confusing, do the authors mean to say that: Black people had the largest average Covid-19 death RQ, followed by AIAN”.

However, in Figure 2 there are states with higher death RQ for AIAN compared to black people. This should be mentioned in the results.

Reviewer 2 Report

The aim of this study was to evaluate national and state-level racial and ethnic disparities in COVID-19 cases, outcomes, and identify potential factors that contribute to any disparities. 

The article is well written and provides valuable data related to the current Covid19 pandemic. The content of the introduction is good. Presentation of the results is clear.

As the authors discuss the details of the pandemic spread, the urban areas the subway usage or personal transportation use, all affect the number of cases and the disparities in a community. These disparities are self generating and represent a cascade but this was not addressed.

More observations could be derived from a more detailed map, such as counties not states, which would not cover the huge differences between urban and rural areas as well as population densities in the eastern or western seaboard vs the Midwest and the South.

Individual and community responses cannot be overlooked during this novel pandemic.

Unfortunately, no policies or resources would be successful based on state-level observations.

Round 2

Reviewer 1 Report

I am happy with the changes the authors made to the study. 

Author Response

Thank you.